# Current and Potential Future Global Distribution of the Raisin Moth *Cadra figulilella* (Lepidoptera: Pyralidae) under Two Different Climate Change Scenarios

**DOI:** 10.3390/biology12030435

**Published:** 2023-03-11

**Authors:** Bing-Xin Wang, Liang Zhu, Gang Ma, Adriana Najar-Rodriguez, Jin-Ping Zhang, Feng Zhang, Gonzalo A. Avila, Chun-Sen Ma

**Affiliations:** 1School of Life Science, Institutes of Life Science and Green Development, Hebei University, Baoding 071002, China; bingxin.wang@wur.nl; 2Climate Change Biology Research Group, State Key Laboratory for Biology of Plant Diseases and Insect Pests, Institute of Plant Protection, Chinese Academy of Agricultural Sciences, Beijing 100193, China; zhuliang01@caas.cn (L.Z.); magang@caas.cn (G.M.); 3Wildlife Ecology and Conservation Group, Wageningen University & Research Centre, Droevendaalsesteeg 3a, 6708 PB Wageningen, The Netherlands; 4The New Zealand Institute for Plant and Food Research, Canterbury Agriculture and Science Center, Lincoln 7608, New Zealand; adriana.najar-rodriguez@plantandfood.co.nz; 5CABI East & South-East Asia, 12 Zhonggunancun Nandajie, Beijing 100081, China; j.zhang@cabi.org (J.-P.Z.); f.zhang@cabi.org (F.Z.); 6MARA-CABI Joint Laboratory for Biosafety, Institute of Plant Protection, Chinese Academy of Agricultural Sciences, Beijing 100193, China; 7The New Zealand Institute for Plant and Food Research Limited, Auckland Mail Centre, Private Bag 92169, Auckland 1025, New Zealand; gonzalo.avila@plantandfood.co.nz

**Keywords:** species distribution model, suitability analysis, CLIMEX, MaxEnt, bioclimatic modeling

## Abstract

**Simple Summary:**

The global trade system is contributing to the spread of invasive species, like the raisin moth (*Cadra figulilella*), causing significant damage to agriculture and the environment, as well as stored food products. The potential distribution of the raisin moth may become even more widespread due to climate change. We newly assessed the potential distribution of the raisin moth globally, under current and future climate scenarios using a climate dataset projected with 23 climate models under two emissions scenarios, and using both CLIMEX and MaxEnt niche modeling tools. Our results indicated that the area of suitable distribution for the raisin moth could increase by 5.24 to 36.37% by the end of the century. Analysis of single predictor showed that excessive precipitation and a temperature range of 0–18 °C during the wettest quarter of the year may impact the species’ establishment. The study highlights the need for using a combined modeling approach, such as CLIMEX and MaxEnt, in future research and the results could be used to inform international trade decisions and environmental risk assessments.

**Abstract:**

Global trade facilitates the introduction of invasive species that can cause irreversible damage to agriculture and the environment, as well as stored food products. The raisin moth (*Cadra figulilella*) is an invasive pest that poses a significant threat to fruits and dried foods. Climate change may exacerbate this threat by expanding moth’s distribution to new areas. In this study, we used CLIMEX and MaxEnt niche modeling tools to assess the potential global distribution of the raisin moth under current and future climate change scenarios. Our models projected that the area of suitable distribution for the raisin moth could increase by up to 36.37% by the end of this century under high emission scenario. We also found that excessive precipitation decreased the probability of raisin moth establishment and that the optimum temperature range for the species during the wettest quarter of the year was 0–18 °C. These findings highlight the need for future research to utilize a combined modeling approach to predict the distribution of the raisin moth under current and future climate conditions more accurately. Our results could be used for environmental risk assessments, as well as to inform international trade decisions and negotiations on phytosanitary measures with regards to this invasive species.

## 1. Introduction

The raisin moth, *Cadra* (=*Ephestia*) *figulilella* (Gregson) (Lepidoptera: Pyralidae), is one of the most economically important worldwide pests of fresh and stored fruits [1]. The raisin moth can attack ripening or overripe fruits (e.g., freshly harvested carobs, dates, figs, grapes) as well as stored dried fruits and nuts (e.g., cashew kernels, cacao beans, apricots kernels, raisins) [1,2,3]. In addition to the damage caused by direct feeding, raisin moth larvae can also cause indirect damage by leaving many crumbs and webs in the fruit, which in combination can negatively affect 50% of fruit production, causing serious economic losses [1]. In some areas, the raisin moth has been reported to cause 90% of fruit damage and contamination, such as in Shadad, Kerman province of Iran [4].

The raisin moth has a relatively short adult life (approx. 16 days) and feeds only during larval form [5]. It undergoes up to 4–5 generations per year [4], and its development rate is greatly affected by temperature and humidity [3]. The thermal biology of the raisin moth was recently summarized by Burks and Johnson [5] and Perring et al. [6]. Under natural conditions, the raisin moth overwinters as fully developed larvae and in the top soil or under the loose bark of the host plants while younger larvae usually do not successfully overwinter [7]. In contrast, all development stages can overwinter in the stored fruit [4]. The longevity of adults is 11–26 days, and they are generally nocturnal and egg-laying. The average number of eggs laid by an adult female is 350, with a maximum of 690 eggs [8]. Several control strategies have been used to control the raisin moth, including cultural control (e.g., early fruit harvesting), physical control (e.g., covering with plastic nets during early ripening season), chemical control (e.g., applying chemical insecticides), reproductive control (e.g., mating disruption), and biological control (e.g., deploying natural enemies like parasitoids, including *Venturia canescens* and *Habrobracon hebetor*) [1,6].

The raisin moth was first recorded as a pest of raisins in California in 1928, and since then, it has spread throughout tropical areas around the world [7]. The raisin moth is likely to be present in areas with hot, dry summers and mild winters, such as Mediterranean, North Africa, Middle East, Middle Asia, and some areas which have similar climates in the Americas and Australia [5,8,9]. In recent decades, with the increase in international fruit trade activities and ongoing global warming, the dispersal of the raisin moth is expected to expand into areas where it has not previously established, including several developing countries and regions that possess favorable climates for fruit production characterized by high temperatures and low humidity. These areas include regions such as Middle Asia, West Asia, basin regions in Northwestern China, and low-lying areas in southwestern China, such as Yuanmou County in Yunnan Province [10,11]. Currently, no efforts have been made in predicting the potential geographic distribution of the raisin moth, under the current and future scenarios of climate change.

Species distribution models (SDMs) are widely used for predicting the past, current, and future species distributions in many ecological, biological and biogeographical studies, especially in the context of climate change. Generally, two types of SDMs, namely correlative approaches and mechanistic approaches are commonly used [12]. Correlative models (e.g., MaxEnt) solely integrate species occurrence data with their environmental variables to forecast species’ potential distribution [13]. In contrast, mechanistic models (e.g., CLIMEX), can incorporate the prior biological knowledge of how species respond to environmental limits from laboratory or field studies [14]. These two types of SDMs have their own advantages and disadvantages [12,15,16]. Correlative models need less information but they have difficulty in predicting novel environmental conditions, while mechanistic models can constrain species ranges by capturing biological knowledge but cannot directly simulate the processes such as which environments are the most favorable for species distribution [12,17]. Both types of SDMs have been widely used to predict potential spreading patterns of invasive species, such as the codling moth (*Cydia pomonella*) [18], the yellow-legged hornet (*Vespa velutina*) [19], and tomato pests (e.g., *Tuta absoluta*) [20].

Here we aimed to (1) assess the global potential distribution of raisin moth under current and future climate scenarios using two different SDMs (i.e., MaxEnt and CLIMEX), and (2) explore the key environmental drivers associated with the potential distribution and suitable habitat changes of the raisin moth. Distribution predictions resulting from this study will provide useful information to help with the developing region-specific quarantine measures for better monitoring, prevention and control of the raisin moth.

## 2. Materials and Methods

### 2.1. Data Collection

#### 2.1.1. Distribution Data

Distribution records of the raisin moth (Figure 1) were compiled from four online databases, namely (1) Global Biodiversity Information Facility, (GBIF, www.gbif.org (accessed on 23 May 2022)), (2) Moth Photographers Group at the Mississippi Entomological Museum (http://mothphotographersgroup.msstate.edu (accessed on 23 May 2022)), (3) Biodiversity Information Serving Our Nation (BISON, https://bison.usgs.gov (accessed on 23 May 2022)), (4) Integrated Digitised Biocollections (iDigBio, www.idigbio.org (accessed on 23 May 2022)) as well as from published studies [4,8,21]. In total, 109 occurrence records were collected. After removing duplicates, erroneous points and records without geographic coordinates, we used a data filtering process by using “spThin” R-package [22] and occurrence data were further reduced to a total of 69 records. We obtained latitude and longitude values from GEOLocate Web Application (see https://www.geo-locate.org/web/WebGeoref.aspx (accessed on 24 May 2022)) for those locations which were not given by the published studies.

#### 2.1.2. Climate Data

Climate data for the CLIMEX model: historical climate data (1951–2020) with the 0.5° × 0.5° resolution were extracted from the monthly gridded Climate Research Unit (CRU, University of East Anglia), Time-series data version 4.0.5 [23]. We extracted five types of meteorological data (monthly average maximum and minimum air temperature (°C), average monthly rainfall (mm) and average relative humidity (%), at both 9 a.m. and 3 p.m., from the dataset to meet the requirements of CLIMEX.

The climate dataset used to run climate change scenarios came from the Coupled Model Intercomparison Project (CMIP) phase 6 models in projections. We extracted the same five groups of meteorological data as above by calculating a multi-model ensemble mean climate dataset, which covers twenty-three Global Climate Models (GCMs) available in the WorldClim database (www.worldclim.org (accessed on 7 June 2022); Table A1). To account for uncertainty in the climate change projections, we presented our results based on the averaged GCM outputs. We chose two shared socioeconomic pathways scenarios (SSP1-2.6 and SSP5-8.5) for two target time periods (mid-century: 2041–2060 and the end of century: 2081–2100), which represented the “best” and the “worst” warming future. SSP1-2.6 represents a low-emission scenario with a warming projection of 2 °C, and SSP5-8.5 represents a high-emission scenario with a warming projection of 5 °C (relative to the years 1880–1900), respectively.

Climate Data for MaxEnt Model: two types of environmental variables were used to parameterize the MaxEnt Model: (1) 19 bioclimatic variables (the average for the years 1970–2000) at 0.16° × 0.16° spatial resolution (∼18 km); (2) topographic variables relative to land morphology, such as elevation (ELEV), aspect (ASPE) and slope (SLOP) (see Table A2 for details). Elevation data (ELEV) was obtained from the WorldClim website (available at https://www.worldclim.org/data/worldclim21.html (accessed on 7 June 2022)) and was used to generate the variables slope (i.e., the incline or steepness of the surface), and aspect (i.e., the compass direction that a topographic slope faces), using the *terrain()* function in “raster” R-package. These variables were selected on the basis of their potential biological relevance to the raisin moth and on their use in previous niche modelling studies of the insect pests [19]. Additionally, we also considered a latitude layer (generated by using the *raster.lat()* function in “red” R-package) as one of the environmental variables, since latitude together with seasonality determined photoperiod [24], which was the common cue to induce insect diapause together with decreased temperatures) [25]. Predicted data for all bioclimatic variables selected were generated by using twenty-three Global Climate Models (GCMs) available in the WorldClim database (www.worldclim.org (accessed on 7 June 2022)) for the two selected scenarios, SSP1-2.6 and SSP5-8.5 in the two time periods, 2041–2060 and 2081–2100.

To assess the multicollinearity between all pairs of environmental variables, we firstly run MaxEnt model with all the environmental variables 10 times to obtain the contribution of each environmental variable, which was used as the criteria for variable selection. Next, we calculated a Pearson correlation coefficient (r) using the *findCorrelation()* function in the “caret” R-package. Variables with high correlation coefficient (|r| > 0.8) and both a low biological relevance and a low contribution to explaining the distribution of raisin moth were removed from the model. As a result, a total of twelve variables were selected to establish the final model: bio2, bio4, bio8, bio13, bio15, bio17, bio18, bio19, ELEV, SLOP, ASPE and latitude (see Table A2; Figure A1).

### 2.2. Bioclimatic Modeling

#### 2.2.1. CLIMEX Model

To calculate species distributions, CLIMEX generates monthly and annual growth indices and four annual stress indices (wet, dry, cold and hot) based on temperature, soil moisture, light, diapause indices and other covariables. It further produces an integrative index, the Ecoclimatic Index (EI), to measure climatic suitability for each given species, which ranges from 0–100, where 0 indicates a location where the climate is totally unsuitable for the species whereas 100 is a location where the climate is optimal in every respect [14,26]. The “Compare Locations” function in CLIMEX was then used to estimate the climatic suitability for the raisin moth. The parameter estimation for the CLIMEX model was based on physiological tolerances of the raisin moth from published laboratory studies. The initial values of parameter were obtained from the built-in Mediterranean template based on the CLIMEX User’s Guide [14] and were modified best to fit the model according to the current distribution of the raisin moth.

To better represent the potential distribution of raisin moth populations in agricultural scenarios, we included an irrigation scenario into our model. A top-up irrigation value of up to 1.5 mm day^−1^ all year round was applied in our model [27]. The top-up scenario would first judge the amount of rainfall. If a given week at any location had more than 10.5 mm of rainfall, no irrigation was added. However, if there was less than 10.5 mm, CLIMEX would make up the deficit by adding irrigation up to the 10.5 mm weekly threshold. The composite potential distribution model was built based on an updated version of global irrigation areas provided by Siebert et al. [28]. If a location was irrigated, the EI accounting for irrigation was used; otherwise, the EI accounting only for natural rainfall was used.

Temperature index: lower temperature threshold (DV0) was set to 11 °C based on the previous study showed that none of the eggs hatched at 10 °C or 12.5 °C [3]. According to the results for development and survival of the raisin moth under different temperatures [25,29], we set the lower optimum temperature threshold (DV1) as 15 °C, the upper optimum temperature threshold (DV2) as 30 °C, and the upper temperature threshold as 36 °C.

Moisture index: we used soil moisture (SM), values from the Mediterranean temperate template [14] to fit the current distribution of the raisin moth.

Cold stress: the cold stress temperature was adjusted at 0 °C to match the raisin moth’s coldest distribution record in Southern Sweden where the winter is generally mild with an average temperature above 0 °C [30], whereas the cold stress rate (THCS) was increased from −0.005 to −0.001 week^−1^ to fit the model to the current distribution of the raisin moth in Europe.

Heat stress: the heat stress temperature threshold (TTHS) was set to 36 °C to match the raisin moth’s hottest distribution record in Queensland, Australia. Additionally, heat stress temperature rate (THHS) was set to 0.0001 week^−1^ to better match the known distribution of the raisin moth to match its known distribution in Mediterranean climates.

Dry stress: we used dry stress threshold (SMDS) and dry stress rate (HDS) value from the Mediterranean template [14] to fit the current distribution of the raisin moth.

Wet stress: wet stress threshold was modified to 2.5 based on SM3, whereas the wet stress rate (HWS) was set to 0.0015 based on the parameters provided in the Mediterranean template.

Hot-wet stress: integrating with the Mediterranean template parameters, the hot-wet maximum temperature threshold (TTHW) was set to 23 °C, and hot-wet stress accumulation rate (PHW) was set to 0.075. The hot-wet stress index (MTHW) was increased from 0.5 to 1.35 to fit the actual distribution of the raisin moth in hot-wet states of the United States (e.g., Florida).

The positive degree day sum: the number of degree days required to complete one generation (PDD) were set to 292, since the tested PDD ranged 236–348 degree-days for each generation [31]. All parameter values used are listed in Table 1.

#### 2.2.2. MaxEnt Model

MaxEnt uses presence-only data as an input to predict the potential distribution of a given species, which is one of the most suitable methods for our raisin moth occurrence data, since absence data are not available for raisin moth at global scale. Instead of using the default parameters in MaxEnt Model, we used the “ENMeval” R-package [32] to perform automated runs to optimize two type of model parameters, Regularization Multiplier (RM) and Feature Combination (FC) to maximize predictive ability and to avoid over-fitting. The spatial block approach was used to split the records into four equal groups, of which three were used for training and the other one was used for testing. This is required if one is to conduct an extrapolation in environmental space [32]. We set the RM parameter to 0.5–4 based on the instruction of the “ENMeval” R-package [32], and each interval was set to 0.5, for a total of eight RM parameters (default setting is 1.00) [33]. For FC parameters, the Maxent model offered 5 features: linear (L), quadratic (Q), hinge (H), product (P) and threshold (T) [34]. We chose 6 feature combinations: L, LQ, H, LQH, LQHP and LQHPT (the default setting is LQHPT). We used the “ENMeval” R-package to evaluate the resulting 48 parameter combinations above. The Akaike information criterion correction (AICc; corrected for small sample sizes) was used to evaluate the fit and complexity of the model [35]; the difference between training and testing AUC (AUC.DIFF) and the 10% training omission rate (OR10) were used to assess the degree of the over-fitting of the model [36]. The parameter combination with the minimum value of AICc were selected as our optimal parameter to build our model [36].

After determining the optimal model parameters, we used “dismo” R-package to stimulate the model in 10-fold cross-validation procedure in MaxEnt, and final output was the average of this procedure. A total of 25% occurrence records were extracted randomly and used to test the dataset, and the remaining 75% of the dataset was used as a training dataset [34]. The maximum background points was set to 10,000. All 12 selected environmental variables were used as predictors. Jackknife testing and response curves were used to evaluate contributions of environmental variable and thresholds to define potential suitable areas for the raisin moth worldwide.

Model performance is quantitatively evaluated by using two types of metrics: threshold-dependent and threshold-independent. The Area Under the Receiver Operating Characteristic (ROC) Curve (AUC) is widely in use since it is the threshold independent measure of model performance. The value of the AUC range is 0–1, and as the AUC value increases, the model’s prediction results become more precise. The model performance is considered poor if the AUC value is less than 0.7, reasonable if the AUC is in the range of 0.7–0.9, and outstanding if the AUC is greater than 0.9 [37]. Furthermore, two metrics that relay on a specific threshold were used, namely the omission rate (OR) at the lowest predicted threshold (LPT) or the minimum training presence threshold, and the omission rate at 10% training presence threshold.

## 3. Results

### 3.1. Projected Potential Distribution of the Raisin Moth under Current Climate Conditions

#### 3.1.1. CLIMEX Model

The projected potential global distribution of the raisin moth resulting from the parameters used in our CLIMEX model covered 98.55% of the current global distribution records for this species (i.e., 68 points out of 69 points) (Figure 2a), which indicated that the model’s prediction was accurate and could be used to further predict the future distribution of the raisin moth and potential shifts under climate change. The climatic suitability (Ecoclimatic Index, EI) was highest in Eastern United States, Southern Brazil, Uruguay, Southern Africa and southwestern, central and eastern regions of China. In addition, a few regions in the western and southern areas in Europe showed favorable climatic suitability for the raisin moth.

#### 3.1.2. MaxEnt Model

The model optimization results indicated that the model included linear, quadratic and hinge (LQ) features, RM = 0.5 and had the lowest AICc value, i.e., delta.AICc = 0 (Figure A2a). The average 10% training omission rate (avg.OR10) value obtained, which was 0.1595, was 61.83% lower than those of the MaxEnt model with the default settings (avg.OR10 = 0.4179), (Figure A2c). Thus, RM = 0.5 and FC = LQ were chosen as the optimal model settings, and these model parameters could reduce the over-fitting and the complexity of the MaxEnt model. The average AUC value of 10 times the repeated training sets was 0.94 ± 0.01 and the average AUC value was 0.91 ± 0.07, suggesting the model performance was good. The MaxEnt model results indicated that the high suitable areas were mainly distributed in the Mediterranean coast and the west southeast coast of the United States. The moderate suitable areas were predominantly located in most parts of Europe, east of the United States, Southern China, and some regions of North Africa. The low suitable areas were primarily situated in the east parts of the United States, Eastern Europe, south of Brazil, Paraguay and part of Bolivia, south of China, and also in the Middle East, including Saudi Arabia, Iraq and Iran (Figure 3a).

MaxEnt model revealed that the most influential environmental variables, based on both percentage contribution and permutation importance, were temperature seasonality (bio4), the mean temperature of wettest quarter (bio8), the precipitation of coldest quarter (bio19), elevation, latitude, and precipitation seasonality (bio15) (Table 2). The cumulative effect of these variables, as represented by their percentage contribution and cumulative permutation importance, was substantial, reaching as high as 88.3% and 88.6%, respectively (Table 2). When testing the role of a single variable, bio19, bio4, latitude, bio8, and elevation resulted in the most significant enhancements in both the regularized training gain (AUC values) and test gain, indicating that these variables exerted an important role in forming the current suitable areas for raisin moth distribution (Figure A3). In contrast, when only mean diurnal range of temperature (bio2) was used in the jackknife test, AUC values and test gain were all close to 0 (Figure A3), indicating that bio2 is not an important predictor for suitable areas of raisin moth. The suitable ranges of mean temperature of wettest quarter (bio8; Figure 4b) was 0–18.0 °C for the raisin moth. The probability of raisin moth presence was higher in areas with low precipitation (<500 mm) during the coldest period (Figure 4c) and in areas of low elevation (<1000 m) (Figure 4d). The probability of the raisin moth being present was high between latitudes 18° and 45°, while it was very low at latitudes <−50° (Figure 4e).

### 3.2. Projected Potential Distribution of Raisin Moth under Climate Change Scenarios

#### 3.2.1. CLIMEX Model

CLIMEX projections of potential future global distribution of the raisin moth, under SSP1-2.6 scenarios, showed that the climatic suitability will increase in Europe but slightly decrease in central China, and part of Sub-Saharan Africa (Figure A4 and Figure A5). The predicted global distribution areas under SSP1-2.6 climate change scenario (4.94 × 10_7_ km^2^) are not expected to change much from the global predicted distribution under current climate (4.7 × 10_7_ km^2^) (Figure A4 and Figure A5). However, under the SSP5-8.5 scenario, the northern distribution limit of raisin moth is expected to move northward, with a drastic increase in climatic suitability in Europe and in North America by the middle of this century (Figure 2b and Figure A6). The total area of suitable regions was estimated to increase by approximately 4.92% (i.e., from 
$4.70\times 10^{7}$
 km^2^ at the current climate conditions to 
$4.93\times 10^{7}$
 km^2^ by the middle of this century). Towards the end of this century, the distribution of the raisin moth is predicted to shift further north, even reaching Southern Canada and Northern Europe (Figure 2c and Figure A6). The overall area of suitable regions was estimated to increase by ca. 5.62% by the end of this century (i.e., from 
$4.70\times 10^{7}$
 km^2^ at current climate conditions to 
$4.97\times 10^{7}$
 km^2^).

#### 3.2.2. MaxEnt Model

Under the SSP1-2.6 climatic scenarios by mid-century, the total suitable areas for the raisin moth are predicted to increase 21.66% by the mid-century (i.e., from current 
$4.80\times 10^{7}$
 km^2^ to future 
$5.84\times 10^{7}$
 km^2^) (Figure A7 and Figure A8). By the end of the century, the increase in total suitable areas for raisin moth will be 22.08% in contrast to conditions under the current climate situations (Figure A7 and Figure A8). Under the SSP5-8.5 climatic scenarios, and by the mid-century, the total suitable areas for the raisin moth will increase by 20.63% (i.e., from current 
$4.80\times 10^{7}$
 km^2^ to future 
$5.79\times 10^{7}$
 km^2^) (Figure 3b and Figure A9). By the end of this century, the total suitable areas for the raisin moth are expected to continue to increase by 36.25% (i.e., from current 
$4.80\times 10^{7}$
 km^2^ to future 
$6.54\times 10^{7}$
 km^2^) (Figure 3b and Figure A9).

### 3.3. Global Distribution Prediction and the Dynamics Shift under Two SDMs

Both CLIMEX and MaxEnt models predicted areas to be highly-suitable for the survival of the raisin moth in Europe and in the east coast of United States, as well as around the Mediterranean coast. However, the MaxEnt model predicts a larger number of suitable areas for the establishment of the raisin moths than CLIMEX predictions. Additionally, predictions in some areas, such as South America, are rather different. The northern boundary of the raisin moth distribution was predicted by both models would move northwards. MaxEnt predicted larger suitable areas in Eastern Europe and the Central of Africa than CLIMEX did, while CLIMEX predicted suitable areas to include the southern part of Africa, which MaxEnt did not predict.

## 4. Discussion

Projections of the potential distribution of the raisin moth developed with both CLIMEX and MaxEnt models match the present recognized distribution of this pest both in its indigenous habitats and in areas where it has been introduced. Both models predicted a similar climatic suitability for a number of regions like the Western United States, Southern Europe and some areas in Middle East and Australia, covering almost all the Mediterranean climatic zone. However, there was a mismatch between CLIMEX and MaxEnt model spatial predictions. For instance, CLIMEX projected somewhat larger areas as highly-suitable in Eastern Argentina, Uruguay, South Africa, Botswana, Zimbabwe and Namibia, whereas MaxEnt projected larger suitable areas in Northern Europe, encompassing Norway, Sweden and Finland. These differences may stem from the varying levels of complexity employed during model fitting, disparities in the algorithms utilized by each niche model, or the absence of comprehensive surveys in certain remote regions that are difficult to reach [39,40]. Additionally, these disparities could be attributed to the nature of the information supplied to each model, with CLIMEX utilizing species-specific biological information, such as thermal development data, while MaxEnt did not [33].

The important role of temperature and humidity for the raisin moth was highlighted by both MaxEnt and CLIMEX and they seem to be the dominant factor determining this insect’s distribution. In the MaxEnt model, the response curve of the mean temperature of the wettest quarter implied that high temperature combined with high rainfall decreased the probability of the raisin moth to establish, which was in agreement with available information from previous studies which indicated that temperature and humidity had a combined impact on the life-cycle of the raisin moth [3] and of other congeneric species [41]. For instance, the optimal relative humidity range for raisin moth was between 30–50%, but it rose to between 50–70% when temperature fell below 22.5 °C or rose above 36 °C [3]. The cumulative effects of temperature and humidity also have impacts on the lifespan of other congeneric species, such as the almond moth (*Cadra cautella*) [41]. The current CLIMEX model developed for the raisin moth also indicate hot-wet stress (HWS) limits the distribution of the raisin moth, which is consistent with our MaxEnt results.

According to MaxEnt, temperature seasonality (i.e., temperature variations) was one of the most important limiting climatic factors playing a role in the potential future distribution of the raisin moth. This resuts corresponds with an earlier study showing that the survival rate of larvae of *Cadra figulilella* and *C. calidella* (Lepidoptera: Pyralidae) decrease to 23% due to a 2.5 °C increase in the optimal temperature [3]. For other related species, such as the almond moth [41], a similar pattern was reported with larvae completing their development at 34 °C later than those ones developing under optimal temperatures of 30–32 °C [41]. Precipitation variations, such as precipitation of the coldest quarter, and precipitation seasonality were also shown by our model to contribute the potential distribution of the raisin moth. A previous study also showed that under the optimal temperature of 30 °C, the development rate of the raisin moth and of the related dried fruit moth, *Cadra calidella*, was reduced and the mortality increased when the insects were kept at 20% r.h. compared to when they were kept at a higher humidity (70% r.h.), which supported our result that humidity affects larval development [3].

Latitude, which serves as an indicator of daylight hours, appears to be a crucial element in shaping the raisin moth’s potential future distribution predicted by the MaxEnt model (Figure A10). Without latitude as a predictor, our MaxEnt model predicted much larger areas of suitable regions including central Australia, Argentina, Uruguay, and the Sichuan Province in China. These are areas in which the raisin moth has never been reported and which do not meet the diapause requirements of the raisin moth. Previous studies also showed that daylight exert a large influence on raisin moth occurrence [25] and other closely-related species [42]. For instance, at an optimal temperature of 30 °C, only 33% of raisin moths emerged in continuous light, while the survival rate was high under continuous darkness, reaching 82% [25]. We did not include the diapause index in our CLIMEX model since most of the diapause requirements of the raisin moth are unknown, apart from what was reported by Cox [25]. Since diapause insects can survive under extreme temperatures [43], incorporating diapause might release the restrictions by extreme climate conditions and resulting in the larger prediction of the potential distribution. Elevation was also recongnized as an important factor linked to the prospective range of the raisin moth, as the higher the elevation and latitude, the lower the overall temperatures [44]. Similarly, latitude and elevation have been demonstrated to have a profound effect in predicting spring emergence on a tortricid moth, the codling moth (*Cydia pomonella*), in North America [45].

A large number of previous studies have demonstrated that climate change would shift the species distribution to north or higher latitudes, especially in high-altitude areas [12,17,46]. In our study, the predictions from both models, CLIMEX and MaxEnt, concurred with these studies and have indicated that the overall suitable distribution area of the raisin moth will increase by 5.19–36.25%, respectively, by the end of this century under future climate scenarios, with the northern limit being predicted to shift northwards. The area of suitable regions will increase, indicating that the potential suitable areas for the raisin moth and the risks of this pest invading new areas will also increase. In some high-altitude areas, the suitable areas will increase (e.g., Norway and Sweden), while in some low-altitude areas, the suitable areas will decrease (e.g., Eastern United States). This suggests that the raisin moth should tend to migrate to high-altitude areas, which matches the results of previous studies [47,48].

It should be noted that CLIMEX and MaxEnt modeling outputs, like any other bioclimatic modeling software, have some limitations (e.g., projections are made considering the climatic variables only), with the exception of CLIMEX that also can include biological data of the species and non-climatic factors (like irrigation). As a result, projections of the potential future distribution of the raisin moth derived from our study should be interpreted carefully given some uncertainties in the role that a number of biotic aspects (e.g., species competition [49], habitat type, food source, or the presence of natural enemies [50]) will have in the model algorithms used by both CLIMEX and MaxEnt [51]. For example, interspecific competitions have previously been reported to exist between the almond moth, the Indian meal moth and the raisin moth [52], which might influence the predicted distribution of the raisin moth or of the other two moth species. Host plants type and availability could also affect the models’ predictions. For instance, a previous study showed that the Indian meal moth adults that were fed a bran diet produced the most offspring, while those that were fed walnuts yielded the lowest number of progeny [53]. Host plant availability also has impacts on pest distribution, but we could not use the current host plants’ distribution to restrain our predicted potential distribution, since data for all host plants of the raisin moth is not available. Additionally, host plant distribution will also shift under future climate change scenarios, and the distribution of economic important plants such as almonds, figs, plums [8], date palms [4] or carobs are also influenced by local governments’ planting guidelines.

There are other potential limitations for both CLIMEX and MaxEnt models that may affect projection outputs. For instance, the output of CLIMEX model is mainly validated by the visual comparison with the current known distribution, and there is no standard statistical approach to optimize this so far [39]. The selection of parameters by the CLIMEX model is subjective even if their settings are based on prior physiological knowledge of the target species. MaxEnt model also has its own limitations such as poor performance when extrapolating to new regions, limited number of presence data, sampling bias, background data extent, the multicollinearity of environmental predictors and choice of predictors [54,55]. Thus, the utilization of two modeling approaches and a comparison of their results could mitigate uncertainties associated with the use of a single model and could serve to offset the limitations of each model, thereby enhancing the overall reliability of model predictions. However, in the current scenarios, the substantial discrepancy in predictions of climate change impacts between the models (and also predictions in a number of areas (e.g., South America)), under the current climate, only serves to compound the uncertainty, as it is challenging to determine which one is more reliable. In addition, raisin moth is a well-known major pest of stored dried fruits, which may dwell and inhabit storage areas such as silos or arrive to new areas inside infested fruits, which means that our predicted future distributions of this pest could become wider than the predictions of our models.

## 5. Conclusions

Here, we report our combined results using both MaxEnt and CLIMEX models to predict the potential distribution of raisin moth under current climate conditions and the distribution shift in response to climate change. Our results show similarities between the two models but also differences, which indicate that future studies should consider using both models together rather than separately. The combined outputs of these two models could serve as a helpful tool for national plant protection organizations to conduct pest risk assessments, and for policy makers and trade negotiators when developing phytosanitary and control measures against high-risk quarantine species like the raisin moth.

## Figures and Tables

**Figure 1 biology-12-00435-f001:**
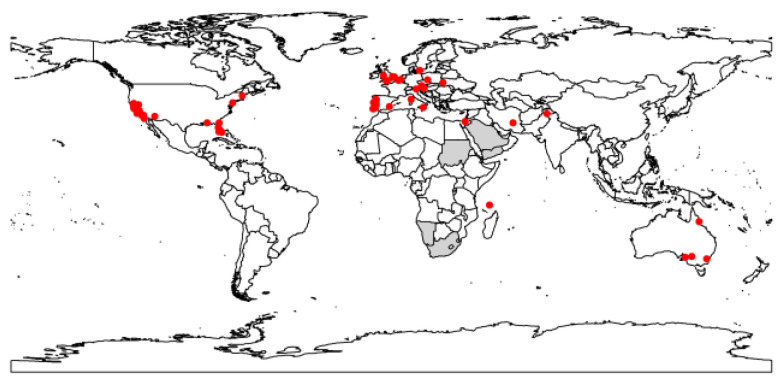
Current worldwide distribution (red dots) of the raisin moth, *Cadra figulilella*. Grey areas represent the distribution of the raisin moth from https://www.afromoths.net/species/show/11662 (accessed on 3 March 2023). These areas were excluded in the model fitting process due to insufficient spatial data (i.e., their precise geographic coordinates could not be determined).

**Figure 2 biology-12-00435-f002:**
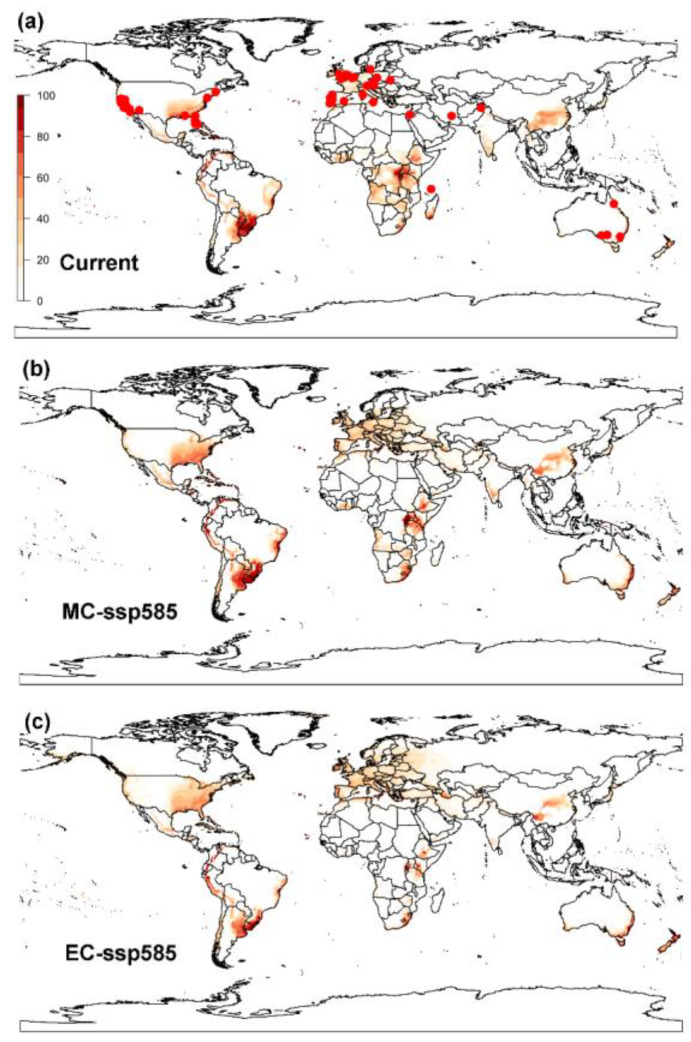
Global potential distribution of the raisin moth, *Cadra figulilella*, predicted by CLIMEX, where (**a**) denotes the potential distribution under current climate conditions, (**b**) denotes the potential distribution under the ssp585 scenarios by mid-century (MC, 2041–2060), (**c**) denotes the potential distribution under the ssp585 scenarios by end of century (EC, 2081–2100). Dark red areas (EI = 100) denote highly suitable areas, while white areas represent unsuitable areas (EI = 0). Red dots denote the current worldwide distribution of the raisin moth.

**Figure 3 biology-12-00435-f003:**
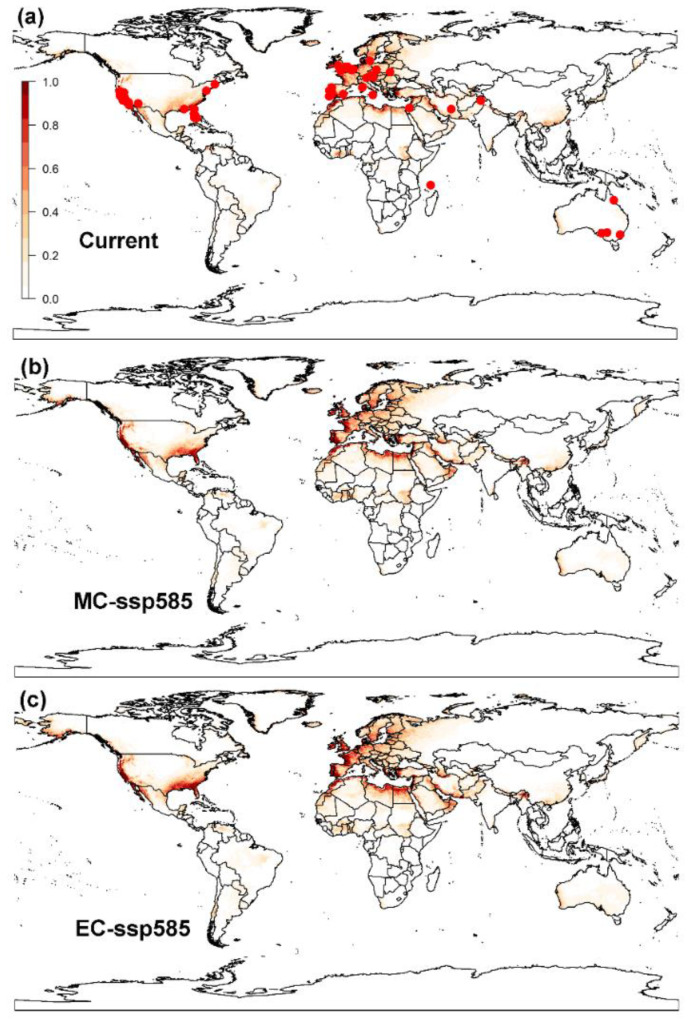
Global potential distribution of the raisin moth, *Cadra figulilella*, predicted by MaxEnt, where (**a**) denotes the potential distribution under current climate conditions, (**b**) denotes the potential distribution under the ssp585 scenarios by mid-century (MC, 2041–2060), (**c**) denotes the potential distribution under the ssp585 scenarios by end of century (EC, 2081–2100). Dark red areas denote highly suitable areas, while white areas represent unsuitable areas. Red dots denote the current worldwide distribution of the raisin moth.

**Figure 4 biology-12-00435-f004:**
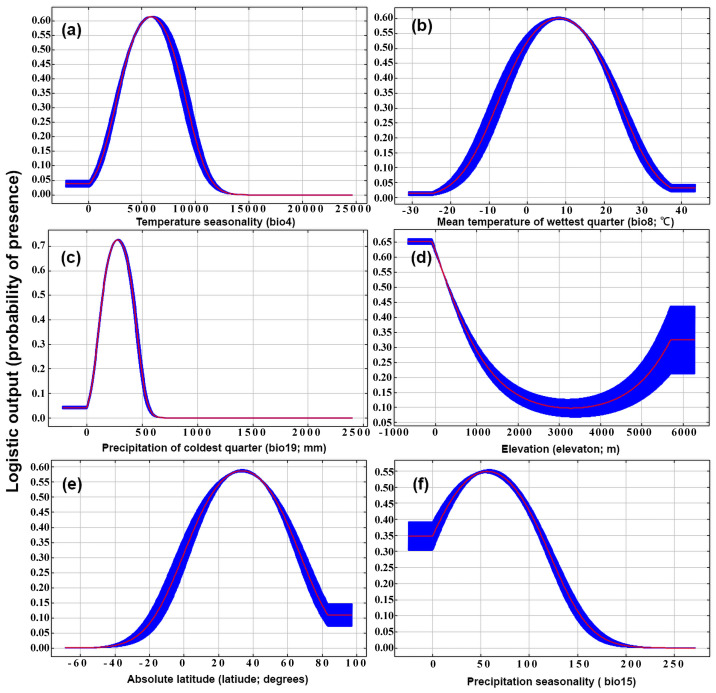
Responses curves of the six important environmental variables for the raisin moth, *Cadra figulilella*; (**a**) temperature seasonality (bio4; i.e., a measure of the temperature change over the course of the year), (**b**) mean temperature of wettest quarter (bio8; °C), (**c**) precipitation of coldest quarter (bio19; mm), (**d**) elevation (elev; m), (**e**) absolute latitude (latitude; °), (**f**) precipitation seasonality (bio15; i.e., an index used to quantify the fluctuation in monthly precipitation levels throughout the year).

**Table 1 biology-12-00435-t001:** CLIMEX parameter values used to fit the potential distribution of *Cadra figulilella*.

Parameters	Descriptions	*C. figulilella*
Temperature		
DV0	Lower temperature threshold (°C)	13
DV1	Lower optimum temperature (°C)	15
DV2	Upper optimum temperature (°C)	30
DV3	Upper temperature threshold (°C)	36
Moisture		
SM0	Lower soil moisture threshold (°C)	0.25
SM1	Lower optimal soil moisture (°C)	0.8
SM2	Upper optimal soil moisture (°C)	1.5
SM3	Upper soil moisture threshold (°C)	2.5
Cold stress		
TTCS	Cold stress temperature threshold (°C)	0
THCS	Cold stress temperature rate (week^−1^)	−0.001
Heat stress		
TTHS	Heat stress temperature threshold (°C)	36
THHS	Heat stress temperature rate (week^−1^)	0.0001
Dry stress		
SMDS	Dry stress threshold	0.02
HDS	Dry stress rate (week^−1^)	−0.05
Wet stress		
SMWS	Wet stress threshold	2.5
HWS	Wet stress rate (week^−1^)	0.0015
Hot-Wet stress		
TTHW	Hot-wet maximum temperature threshold (°C)	23
MTHW	Hot-wet moisture threshold	1.35
PHW	Hot-wet stress accumulation rate (week^−1^)	0.075
PDD	Effective accumulated temperature (degree-days)	292

Footnote: Values without units are dimensionless indices. Definitions of all CLIMEX parameters included in this table are described in Kriticos et al. [14].

**Table 2 biology-12-00435-t002:** Percentage contribution and permutation importance values for each environmental variable included in a MaxEnt model used to predict current and future worldwide distributions of the raisin moth, *Cadra figulilella*.

Variable	Descriptions	Percent Contribution	Permutation Importance
bio4	Temperature Seasonality (standard deviation ×100)	23.3	37.1
bio8	Mean Temperature of Wettest Quarter	18.3	1.6
bio19	Precipitation of Coldest Quarter	17.7	32.8
elev	Elevation	14	5
latitude	Latitude	12	9.6
bio15	Precipitation Seasonality (Coefficient of Variation)	3	2.5
bio17	Precipitation of Driest Quarter	2.8	0.5
bio2	Mean Diurnal Range (Mean of monthly (max temp–min temp))	2.6	0.7
aspect	Aspect	2.2	0.4
bio13	Precipitation of Wettest Month	2.2	4.7
slope	Slope	1	2
bio18	Precipitation of Warmest Quarter	0.9	3.1

Note: bio4 represents the variation in temperature throughout the year, which is calculated by determining the standard deviation of the mean monthly temperatures for the 12 months. A higher standard deviation indicates a greater range of temperature variability [38]. bio15 is an index used to measure the variability in monthly precipitation levels throughout the year. It is expressed ad a percentage and is calculated by dividing the standard deviation of the monthly total precipitation by the mean monthly total precipitation (also known as the coefficient of variation) and is expressed as a percentage [38].

## Data Availability

The global gridded climate data and climate projections are available from WorldClim (http://www.worldclim.org (accessed on 7 June 2022)) and the monthly gridded Climate Research Unit (CRU, University of East Anglia). Distribution records of the raisin moth are available from four online databases: (1) Global Biodiversity Information Facility, (GBIF, www.gbif.org (accessed on 23 May 2022)), (2) Moth Photographers Group at the Mississippi Entomological Museum (http://mothphotographersgroup.msstate.edu/ (accessed on 23 May 2022)), (3) Biodiversity Information Serving Our Nation (BISON, https://bison.usgs.gov (accessed on 23 May 2022)), (4) Integrated Digitised Biocollections (iDigBio, www.idigbio.org (accessed on 23 May 2022)) as well as from published studies (denoted in the main text).

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
