# Peer review of "Current and Potential Future Global Distribution of the Raisin Moth Cadra figulilella (Lepidoptera: Pyralidae) under Two Different Climate Change Scenarios"

_biology, 2023, doi:10.3390/biology12030435_

Round 1
Reviewer 1 Report
Comments and Suggestions
The research aimed to put forth the global trade system and its connection with the transfer of invasive pests from one region to another. Overall, the manuscript is well written and, importantly, shows the current worldwide distribution of the raisin moth and the possible increase in population in a decade. Also, the combined model studies reveal the climatic changes and impact on the raisin moth populations. However, I have a few questions and suggestions that can be incorporated into the manuscript to enhance the overall quality of this manuscript.
Questions
1. Line 200 - "The spatial block approach was used to divide the records into 4 equal groups, of which three were used for training and the other one was used for testing". This approach used 4 equal groups: three were trained, and one was tested. Is there any particular reason? Why did the author not try two training and two tests?
2. Is any statistical approach applicable to these combined model studies for sampling or data acquisition?
Suggestions
1. In the introductions section - The authors can add one or two lines about the current control measure of the raisin moth in its native region. This information could link with pest control strategies and lead the researchers who referred to the article in the future.
2. Page 18 – Figures 1, 2, 3 – the labeling inside the figures looks too small. Even though I zoomed the PDF file to 400%, the axis title and numbers still look tiny. I suggest you edit the figure for good appeal.
Reviewer 2 Report
In generall it is well prepared manuscript. I have only few minor suggestions (see manuscript).
Main question is about not complete distributional data of Cadra figulilella used in the analysis, i.e. available distribution data from Sub-Saharan Africa (https://www.afromoths.net/species/show/11662) are not included here (see Figure 1).
